# Evaluation of the Effectiveness of Aeration and Chlorination during Washing to Reduce *E. coli* O157:H7, *Salmonella enterica*, and *L. innocua* on Cucumbers and Bell Peppers

**DOI:** 10.3390/foods13010146

**Published:** 2023-12-31

**Authors:** Julysa A. Benitez, Jyoti Aryal, Ivannova Lituma, Juan Moreira, Achyut Adhikari

**Affiliations:** School of Nutrition and Food Sciences, Louisiana State University AgCenter, 261 Knapp Hall, Baton Rouge, LA 70803, USA; abrilbz1994@tamu.edu (J.A.B.); ajyoti1@lsu.edu (J.A.); ilituma1@lsu.edu (I.L.); jmoreiracalix1@lsu.edu (J.M.)

**Keywords:** forced aeration, chlorine, *Listeria innocua*, *Salmonella enterica*, *E. coli* O157:H7

## Abstract

The attachment strength of bacteria to surfaces can affect the efficacy of sanitizers during washing. This study aimed to determine the effectiveness of chlorination and aeration in the removal of pathogens from the surface of produce. Cucumbers and bell peppers were inoculated with *Listeria innocua*, *Escherichia coli* O157:H7, or *Salmonella enterica*; afterwards, the produce was washed with or without chlorinated water (100 ppm) for 3 min in combination with or without aeration. Cucumbers washed with chlorinated water, with or without aeration, presented significant reductions of *L. innocua* (3.65 log CFU/cm^2^ and 1.13 log CFU/cm^2^, respectively) (*p* < 0.05). Similarly, bell peppers washed in chlorinated water with aeration (1.91 log CFU/g) and without aeration (2.49 log CFU/g) presented significant reductions of *L. innocua*. A significant reduction of *L. innocua* was observed on bell peppers washed with non-chlorinated water with aeration (2.49 log CFU/g) (*p* < 0.05). Non-chlorinated water was also effective in significantly reducing the level of *Salmonella enterica* (*p* < 0.05) on cucumbers and bell peppers. Washing with chlorinated water with aeration reduced *Salmonella enterica* levels from 4.45 log CFU/cm^2^ on cucumbers to below the detectable limit (0.16 log CFU/cm^2^). The highest reduction of *Salmonella enterica* from bell peppers occurred after washing with chlorinated water with aeration (2.48 log CFU/g). *E. coli* O157:H7, *L. innocua*, and *Salmonella enterica* levels present in non-chlorinated water after washing contaminated produce with or without aeration were significantly greater than those in chlorinated water (*p* < 0.05). After treatment, the population levels of all pathogens in chlorinated water with or without aeration were below the detectable limit for bell peppers (<1.10 log CFU/mL) and cucumbers (<1.20 log CFU/mL). Using chlorine in combination with forced aeration during washing minimizes cross-contamination of bacterial pathogens.

## 1. Introduction

Fresh produce is mainly consumed with minimal processing and is associated with several foodborne illness outbreaks [1]. Cross-contamination of fresh produce represents a significant concern for food safety, with contamination occurring at any point of production (cultivation, harvesting, processing, transportation, and retail sales) [2,3,4]. Numerous pathogens, including Shiga-toxin-producing *Escherichia coli* (STEC), *Salmonella* spp., *L. monocytogenes*, and human parasites, have been linked to outbreaks from fresh produce consumption [5]. One of the most recent outbreaks in the United States associated with fresh produce was a *Listeria* outbreak in 2023, linked to leafy greens, that resulted in 19 illnesses and 18 hospitalizations across 16 states [6]. Similarly, a *Salmonella* spp. outbreak in 2022 linked with alfalfa sprouts resulted in 63 illnesses and 10 hospitalizations in 8 states [7]. Additionally, in 2016, a total of 907 people were infected in an outbreak involving strains of *Salmonella* poona, and 519 people reported eating cucumbers. This resulted in 204 people being hospitalized and 6 reported deaths [8]. The post-harvest handling of produce within packaging or processing facilities can be a contributing factor to increased microbial contamination [9].

Washing fresh produce is crucial for the removal of soil and debris and enhancing the appearance of the product [10]. Additionally, washing aids in the reduction of the microbial load present on the surface of fresh produce, improving the quality, shelf-life, and safety of the produce [11]. However, during the washing process, microorganisms loosely adhering to contaminated surfaces can be released into the water, risking cross-contamination of other produce [12]. The use of sanitizers such as peroxyacetic acid, ozone, chlorine, and chlorine dioxide during washing can minimize the risk of cross-contamination [13]. Among these, chlorine stands out as a widely used sanitizing agent due to its low cost and its proven efficiency in the removal of microorganisms from produce and surfaces [14]. However, traditional chlorine treatments have proven to be inconsistent in controlling pathogens in fresh produce [15]. Despite this, commercial chlorine sanitizers are still widely used, with concentrations typically ranging from 50 to 200 ppm [13].

Some pathogenic bacteria can become firmly attached to surfaces, making them very difficult to remove and reducing the effectiveness of sanitizers like chlorine [16]. Once produce surfaces are contaminated, pathogens can not only survive but also significantly increase throughout the shelf-life of these products [17,18]. Microorganisms become highly tolerant to various disinfectants once they are strongly attached to or in biofilms. In these conditions, biofilms are particularly persistent and typically require scrubbing for removal [12]. These findings highlight the need for researchers to investigate novel approaches that can reduce the risk of cross-contamination and minimize bacterial populations, even after bacteria have become attached to surfaces.

Certain studies have focused on developing alternative sanitizing methods by evaluating novel sanitizers or combining two or more agents with physical techniques [15]. These multi-faceted approaches, known as hurdle technology, are commonly used to increase the efficacy of disinfectants [19]. Methods like ultraviolet radiation (UV), radio frequency (RF), ionizing radiation, and ultrasound are often utilized for disinfection; however, they are expensive and commonly prove impractical in small-scale operations [20,21]. In Japan, force aeration has been used along with water to wash surfaces or produce, destroying 90% more microorganisms [22]. The efficacy of forced aeration is attributed to the scrubbing action of the bubbles and the energy released during their collapse, facilitating the better removal of microorganisms from produce surfaces [23,24]. A previous study evaluated the impact of Ultrafine Bubble (UFB) technology (using CO_2_ and air) on the effectiveness of chlorine and peracetic acid solutions during the washing of Gala apples contaminated with *E. coli* O157:H7 and *L. monocytogenes* [25]. The incorporation of CO_2_ UFBs in combination with sanitizers significantly reduced pathogenic populations (2.10 and 2.40 log CFU/apple, respectively), compared to solutions without UFBs. Air UFBs presented similar reductions of *E. coli* O157:H7 and *L. monocytogenes* when combined with sanitizers (1.90 and 2.20 log CFU/apple, respectively) [25]. Additionally, a review of the challenges in improving the microbial food safety of fresh produce and the limitations of post-harvest washing suggests that a combination of interventions can lead to a reduction in produce contamination. These interventions include post-harvest washing, which could be applied as a hurdle for the control of contamination by foodborne pathogens [26]. Furthermore, research conducted on forced aeration bubbles to control *L. monocytogenes* and *Salmonella* Newport from previously contaminated Roma tomatoes and cantaloupe determined that the combination of forced airflow and chlorine in a water container increased the removal of these pathogens [27].

This creates a basis for further research to determine if the combination of physical methods (i.e., aeration) and chemical methods (i.e., sanitizer solutions) can be an effective alternative for the disinfection of pathogens on the surfaces of fruits and vegetables. Bubbles can be generated at different diameters and velocities, and their effectiveness might vary depending on the pathogen and food of interest [27]. Moreover, for this method to be widely adopted, especially by small and medium-scale industries, the forced aeration system must be designed to be cost-effective. Most sanitizers used commercially employ chemical cleaning agents and may not incorporate brushes or rollers during sanitization [28]. Thus, the objective of this study was to evaluate the effectiveness of aeration and the effect of chlorination during washing to dislodge microorganisms from cucumbers and bell peppers.

## 2. Materials and Methods

### 2.1. Inoculum Preparation

*L. innocua* strain ATCC 33090, a cocktail of five strains of *E. coli* O157:H7: Odwalla strain 223, F4546, EC 4042, H1730, ATCC 43895, and a cocktail of three strains of *Salmonella enterica* Enteritidis PT30, *Salmonella enterica* Tennessee strain K4643, and *Salmonella enterica* ATCC 14028 were used for this study. The frozen cultures were thawed inside a biosafety cabinet (LabGard NuAire Inc., Plymouth, MN, USA). Cultures were activated by transferring 0.1 mL of the thawed culture to a separate 10 mL test tube of Tryptic Soy Broth (TSB) (Hardy Diagnostics, Santa Maria, CA, USA) with 0.6% yeast extract (TSBYE) (VWR, Radnor, PA, USA) for *L. innocua* and TSB for *Salmonella enterica* and *E. coli* O157:H7 strains, which were later vortexed. All the inoculated test tubes of TSB or TSBYE were incubated at 37 °C for 24 h. All media were previously supplemented with 50 µg/mL of nalidixic acid. After the incubation period was completed, the inoculated test tubes were removed from the incubator and placed inside a biosafety cabinet. Each test tube was then vortexed, and 0.1 mL of its contents were transferred to new sterile 10 mL test tubes of TSB or TSBYE, followed by an incubation period at 37 °C for 24 h. This activation procedure was carried out once more after the completion of the 24 h. Upon completion of the 72 h of the procedure, microbial cells were harvested by centrifuging at 6026 *g* for 10 min at 4 °C. Afterwards, the supernatant was removed and the pellet was rehydrated with Phosphate Buffer Saline 1X (PBS) (VWR, Radnor, PA, USA). The individual strains of each specific microorganism were combined to form a cocktail, with each cocktail having a concentration of approximately 8 log CFU/mL. 

### 2.2. Inoculation of Produce

Unbruised cucumbers (*Cucumis sativus)* and bell peppers (*Capsicum annuum*) were obtained from the local supermarket on the day of the experiment. The bell peppers and cucumbers were placed in a biosafety hood and were spot inoculated with *L. innocua*, *E. coli* O157:H7, and *Salmonella enterica*. Aliquots of 200 µL (8 log CFU/mL) were used for each bell pepper and cucumber, similar to previous studies [12,17,18]. Around 40 droplets were inoculated on the surface of the bell pepper and two droplets around the stem, and 40 droplets were inoculated on one side of the cucumber. After inoculation, bell peppers and cucumbers were left to dry for 1 h at room temperature inside a biosafety cabinet.

### 2.3. Aeration System Design

The system was made with 4 pieces of 3.16 cm × 30.48 cm PVC pipes joined together in a rectangle as shown in Figure 1. This structure was linked to a vertically aligned pipe of the same dimensions, which was in turn connected to a blower boasting an airflow of 161.4 m³/h and a velocity of 55.88 m/s. Each tube within the rectangle had perforations of 0.50 cm in diameter at a one-inch gap, which allowed the generation of bubbles when the air was introduced through the blower connected to the pipe. The mode of action of this apparatus consisted of the generation of bubbles through these perforations via the airflow of a blower. The rectangular assembly was positioned at the bottom of the bucket/tank, fully immersed in the wash solution.

### 2.4. Chlorine Treatment

A total chlorine solution of 100 ppm was prepared by adding 5% sodium hypochlorite (Clorox, Oakland, CA, USA) to sterile water, and 0.1 N of citric acid was added to maintain the pH of the chlorine solution at 7. Chlorine concentration was measured using diethyl-p-phenylenediamine (DPD), a colorimetric reaction determined by a Hach DR 900 Multiparameter Colorimeter (Hach, Loveland, CO, USA) and Clorox Smart Strips (Clorox, Oakland, CA, USA). Additionally, the pH of the solution was measured using a VWR H30PC0 Multi-Parameter Handheld Meter (VWR, Radnor, PA, USA). Twelve liters of water were used for each treatment. After inoculating and drying the produce inside a biosafety cabinet for 1 h, the inoculated cucumbers and bell peppers were washed with non-chlorinated water or chlorinated water at a concentration of 100 ppm for 3 min with or without the application of aeration. Treatments consisted of combinations of washing (non-chlorinated water or chlorinated water) and forced aeration (with or without bubbles), with control samples consisting of produce being inoculated and then sampled (without any wash or aeration application). During each replicate, four duplicates of the respective produce made up each treatment and the control samples. The bell pepper trials were replicated three separate times, and the cucumber trials were replicated four different times.

### 2.5. Sample Analysis

Each bell pepper sample was transferred to stomacher bags containing 100 mL of PBS 1X immediately after washing. Bell peppers were homogenized along with PBS 1X using a Bagmixer^®^ 400 blender (Interscience Laboratories Inc., Woburn, MA, USA). The formed homogenate was considered the initial dilution based on the added volume of PBS and the weight of the bell pepper. Further dilutions were made by taking 1 mL of the homogenate and serially diluting it using 9 mL test tubes containing PBS 1X. Whole cucumber samples were placed in sterile bags containing 200 mL of PBS 1X. A higher volume of this diluent was used due to cucumber samples being longer than bell pepper samples and the additional volume being sufficient to appropriately massage cucumbers for dislodging bacteria. These cucumber samples were hand massaged for 3 min, in order to rinse bacterial cells, with the initial dilution being determined by the added volume of PBS 1X and the weight of the cucumber. After massaging to form the first dilution, further dilutions of the cucumber sample were made by diluting 1 mL of the initial dilution accordingly in 9 mL test tubes containing PBS 1X. The samples were then plated using an inoculating volume of 0.1 mL of Xylose-lysine Deoxycholate Agar (XLD) (Hardy Diagnostics, Santa Maria, CA, USA) for *Salmonella enterica*, Sorbitol-MacConkey Agar (SMAC Agar) (Hardy Diagnostics, Santa Maria, CA, USA) for *E. coli* O157:H7, and Oxford *Listeria* Agar (Hardy Diagnostics, Santa Maria, CA, USA) for *L. innocua*, with all agars being supplemented with 50 µg/mL of nalidixic acid (VWR, Radnor, PA, USA) during preparation. The plates were then incubated at 37 °C for 24 h in a Sanyo MCO-18AIC (UV) CO_2_ Incubator (SANYO Electric Co., Ltd., Osaka, Japan). Additionally, a 10 mL sample of the wash solution, both non-chlorinated and chlorinated water from all treatments, was taken immediately after taking either the cucumbers or bell peppers to detect the bacterial load left in the wash solution. From these 10 mL samples, 1 mL was removed and later separated into three separate subsamples (300 µL, 300 µL, and 400 µL). These were each spread-plated on separate Petri dishes containing the respective agars for either *Salmonella enterica*, *E. coli* O157:H7, or *L. innocua.* The plates were incubated at 37 °C for 24 h. Colony-forming units (CFU) were counted after 24 h and converted to log CFU/g for bell peppers and log CFU/cm^2^ for cucumbers. The average weight of bell pepper samples was 191.20 ± 17.41 g, and for cucumber samples, the average weight was 244.85 ± 26.66 g.

### 2.6. Statistical Analysis

The data were analyzed using the Statistical Analysis Software (SAS Institute, Cary, NC, USA) through an ANOVA test to identify if significant differences existed between treatments. In this study, cucumbers and bell peppers underwent four treatment variations: non-chlorinated water with aeration, non-chlorinated water without aeration, chlorinated water with aeration, and chlorinated water without aeration. Upon establishing significant differences between these treatments, a Tukey post-ANOVA test was applied to determine mean differences, with a significance level set at *p* < 0.05.

## 3. Results and Discussion

### 3.1. Efficacy of Aeration during Washing of Cucumbers and Bell Peppers against L. innocua

Washing with non-chlorinated water was not effective in reducing the level of *L. innocua* significantly (*p* > 0.05) in cucumbers and bell peppers (Figure 2 and Figure 3). A significant reduction (*p* < 0.05) of *L. innocua* was achieved when washing cucumbers in chlorinated water without aeration (1.13 log CFU/cm^2^) and with aeration (3.65 log CFU/cm^2^) as compared to washing with non-chlorinated water. There was no significant reduction (*p* > 0.05) of *L. innocua* when washing bell peppers with chlorinated water or non-chlorinated water with or without aeration.

The morphology of fresh produce can affect its susceptibility to contamination. This can be categorized into three categories: product shape, macrostructures (stem bowl, calyx), and microstructures (wax, trichomes, microcracks, lenticels, and stomata) [29]. An irregular shape may reduce the effectiveness of washing by limiting contact with scrubbing brushes and sanitizers. The presence of macrostructures and microstructures may help bacteria attach more easily through physical entrapment [29].

Differences between fruits and vegetables and the location of bacteria contamination (external or inner surface) can lead to lower effectiveness for sanitizers used in the decontamination of fresh fruits and vegetables [30,31]. For instance, a study that focused on the location of *E. coli* O157:H7 on the outer and inner surfaces of apples affected by bruising, washing, and rubbing concluded that *E. coli* O157:H7 was mainly attached to sites of the apple that would protect the bacteria against removal, such as wax platelets and lenticels on the surface of the fruit [32]. In our study, the reduction of *L. innocua* on bell peppers was not significant when washing with either water alone or chlorinated water, regardless of the use of aeration. Bacteria are capable of attaching firmly to the microstructures present on bell peppers, making it difficult to effectively clean and disinfect this product [33]. Similar results were found in apples, where *L. innocua* was attached primarily to the apple peel in the stem bowl and calyx sections and embedded in the microcracks [33].

In the case of cucumbers, a significant reduction (*p* < 0.05) of *L. innocua* was observed when cucumbers were washed with chlorinated water with and without aeration. As mentioned, the surface characteristics of the produce play a role in bacterial attachment and removal [34,35]. The inoculated cucumbers had a relatively smooth surface and did not have a stem or calyx section; this potentially indicates that the use of a sanitizer with or without bubbles may be sufficient to remove *L. innocua* from the surface of this produce. A previous study reported that 100 ppm of chlorinated water was effective in reducing the population of *L. monocytogenes* in cucumbers and tomatoes [36]. Another study involving the intentional introduction of bacteria to fresh tomatoes revealed that immersing them in a 200 ppm chlorine solution for 15 min led to an 8.06 log reduction of *E. coli* from the tomato surfaces, while dipping lettuce leaves in a 100 ppm chlorine solution resulted in a 3.00 log reduction of *Salmonella* spp. [37]. In our study, 100 ppm of chlorinated water was effective in reducing the population of *L. innocua* in cucumbers by 1.74 log CFU/cm^2^ and 0.96 log CFU/g in bell peppers.

### 3.2. Efficacy of Aeration during Washing of Cucumbers and Bell Peppers against E. coli O157:H7

Washing with non-chlorinated water was effective in reducing the level of *E. coli* O157:H7 significantly (*p* < 0.05) in cucumbers and bell peppers (Figure 4 and Figure 5). A significant reduction of *E. coli* O157:H7 (2.86 log CFU/cm^2^) was observed on cucumbers treated with chlorinated water with aeration. Significant reductions were observed on bell peppers (*p* < 0.05) when these were washed with non-chlorinated water with aeration and chlorinated water with or without aeration.

Previous studies have shown that the surface roughness of produce can significantly reduce the efficiency of sanitation methods. For instance, the results of a study by Zhang and Tikekar (2021) on the effectiveness of air microbubbles on the detachment and inactivation of pathogens on grape tomatoes, blueberries, and baby spinach showed that the attachment was stronger on spinach compared to grape tomatoes [38]. This same study concluded that the roughness of the surface of baby spinach allowed the entrapment of bacteria, making it difficult to remove [38]. It is also important to note that the mechanisms behind bacterial attachment to produce surfaces are not only related to the surface structure and biochemical characteristics of the produce but also to the specific bacteria [39]. Studies have shown that high-surface wax vegetables such as cucumbers and bell peppers differ greatly from the rough surfaces of leafy greens and allow more bacterial removal than low-surface wax produce [40]. Additionally, *E. coli* O157:H7 has shown a preferential affinity to cut surfaces of produce and leafy greens rather than uncut intact surfaces [15,41,42], which could explain why washing with non-chlorinated water and without aeration was effective in reducing *E. coli* O157:H7 in both bell peppers and cucumbers.

### 3.3. Efficacy of Aeration during the Washing of Cucumbers and Bell Peppers against Salmonella enterica

Washing with non-chlorinated water was effective in reducing the level of *Salmonella enterica* significantly (*p* < 0.05) in cucumbers and bell peppers (Figure 6 and Figure 7). Washing with chlorinated water with aeration was able to completely reduce the initial 4.45 log CFU/cm^2^ of *Salmonella enterica* on cucumbers to below the detection limit of 0.16 log CFU/cm^2^. The most significant reduction of *Salmonella enterica* on bell peppers was observed when washing with chlorinated water with aeration (2.48 log CFU/g).

Washing *Salmonella enterica* off the surfaces of bell peppers and cucumbers was efficient with non-chlorinated water. However, the greatest reduction was achieved by using chlorinated water with aeration. Studies suggested that the inactivation of *Salmonella* spp. in fresh produce was mostly influenced by the produce type [16]. Produce with smooth surfaces, such as bell peppers and cucumbers, provided very little protection for *Salmonella* spp. from chemical treatments, leading to increased effectiveness in the removal of this pathogen [16]. A study conducted by Ding et al., (2015) concluded that the variation in effectiveness of decontamination was attributed to the topographical features of the produce. A greater decrease in microbial load was noted in smooth-surfaced cherry tomatoes compared to strawberries with an uneven surface [43]. Factors such as target microorganisms and characteristics of the produce surface will influence the success of the washing of fresh produce [44]. This is consistent with the results of a study by Li et al., (2020) that compared the efficacy of different washing procedures on cucumber and tomatoes. Findings from this study suggested that the antimicrobial efficacy of chlorinated water at a concentration of 100 ppm significantly removed *Salmonella* spp., regardless of the produce type [36].

### 3.4. Level of Contamination of L. innocua, E. coli O157:H7, and Salmonella enterica in Wash Water and Chlorine Solution

*L. innocua, E. coli* O157:H7, and *Salmonella enterica* levels in non-chlorinated wash water with aeration and without aeration were significantly higher (*p* < 0.05) than chlorinated wash water (Table 1). *L. innocua*, *E. coli* O157:H7, and *Salmonella enterica* levels in chlorine wash solutions with or without aeration were below the detectable limit of the test (<1.10 log CFU/mL for bell pepper wash water and <1.20 log CFU/mL for cucumber wash water).

The level of contamination in the water after the produce was treated was much higher in non-chlorinated wash water than in chlorinated water, regardless of the use of aeration. This suggests that chlorine is effective in reducing microbial populations and preventing cross-contamination as the most widely used sanitizer [45]. However, high chlorine dosages are usually the most common approach, even though the efficiency of antimicrobial treatments used can also be affected by factors such as the organic matter content present in the water and the type of produce being washed [46]. Nevertheless, as indicated by Lopez-Galvez et al., (2021) the use of chlorine remains a safe alternative, since previous studies using non-chlorinated water have shown that this may increase the risk of cross-contamination [47]. This is consistent with the results of this study, which indicated that after the produce is washed with water, the microorganisms may remain in the wash water and contaminate other produce that was not previously contaminated. However, when chlorine is present, the levels are below the detectable limit of the test. Furthermore, Truchado et al., (2023) reported that *E. coli* O157:H7 cells can be damaged during washing of produce surfaces, becoming viable but non-culturable (VBNC). which may serve as an explanation for the population levels of *E. coli* O157:H7 being below the detectable limit of the test in non-chlorinated water with bubbles from cucumbers (<1.20 log CFU/mL) and non-chlorinated water with bubbles from bell peppers (<1.10 log CFU/mL). The damaged cells can attach to other surfaces that come in contact with the wash water; however, VBNC *E. coli* O157:H7 cells are rarely able to resuscitate on the surface of produce [48]. The ability of the bubbles to dislodge bacteria from their surfaces is due to their scrubbing action or turbulence when they collapse, generating free radicals [38]. The bubbles may potentially play a crucial role in removing bacteria from the produce surface, improving the efficacy of sanitizers, and reducing the necessary dosages of sanitizers used to wash produce. Aeration or bubbles have the potential to improve the efficacy of sanitation and minimize the use of chemicals, making them an environmentally friendly technology as well [25,49,50].

It is important to add that the removal of pathogens from produce surfaces is also related to bacterial attachment behavior, which may differ depending on both the surface of the produce and the microorganism as well as the biochemical features of the bacteria [51]. In this study, the inability of bubbles to effectively dislodge *L. innocua* may be due to the biofilm formation that keeps the cells embedded in the produce even after washing [17]. Aeration proved more effective in reducing *E. coli* O157:H7 and *Salmonella enterica*. *E. coli* O157:H7 has been shown to internalize injuries on the surfaces of apples and lettuce rather than intact surfaces [15]. This could be the reason aeration was effective in removing *E. coli* O157:H7 from cucumber surfaces, even when washing with non-chlorinated water. Removal of *Salmonella* spp. from produce surfaces depends on the surface of the produce [52]. For instance, irregularities in the surface of the produce, such as scars and gaps, might provide shelter to *Salmonella* spp. during treatment [53]. *Salmonella* spp. contamination of cucumbers using irrigation water was simulated by Burris et al., (2020) to assess the pathogen’s uptake by the stems. The authors discovered a significant amount of inoculum, 8.30 log CFU/root zone, translocated to the stem, which is consistent with the idea that *Salmonella* spp. can infect cucumber plants and move up the stems [54]. In our study, the intact bell peppers and cucumbers provided no shelter for *Salmonella enterica* during treatment, allowing for the population to be reduced to below detectable limits in the case of chlorine treatments.

*Listeria* spp. and *Salmonella* spp. attach differently to the same surface, with attachment varying according to surface properties as well as the bacteria’s characteristics. According to a study by Sinde & Carballo (2000), *L. monocytogenes* and *Salmonella* spp. adhesion to various surfaces differed greatly. This was attributed mainly to *Salmonella* spp. strains being more hydrophobic and having lower surface free energy than *L. monocytogenes* strains. Additionally, they discovered that, compared to *Salmonella* spp., *L. monocytogenes* adhered to all surfaces in higher quantities [55]. The attachment of *Salmonella* spp. and *L. monocytogenes* to tomatoes at various points in the supply chain was examined by Cabrera-Diaz et al. (2022). They discovered that the pathogens were influenced differently by the storage conditions. *Salmonella* spp. was adversely affected by storage time but not temperature, while *L. monocytogenes* was influenced greatly by storage time, temperature, and relative humidity [56].

## 4. Conclusions

Forced aeration or bubbles have a great potential to facilitate the dislodging of microorganisms from their surfaces. The inactivation of microorganisms depends on the type of produce, their surface characteristics, and the type of microorganism. Produce with rough, leafy surfaces or that contains stem bowls, calyx, wax, trichomes, microcracks, lenticels, and stomata provides a shelter for bacteria attachment, making it more difficult to remove. In this study, washing with non-chlorinated water reduced the microbial population of *E. coli* O157:H7 and *Salmonella enterica* from both cucumbers and bell peppers. However, in the case of *L. innocua*, washing with non-chlorinated water was not able to significantly reduce the bacterial population in either cucumbers or bell peppers since *L. innocua* attaches more firmly to the surface of the produce.

After washing bell peppers and cucumbers (contaminated with *L. innocua, E. coli* O157:H7, and *Salmonella enterica*) with non-chlorinated water with or without aeration, a certain level of contamination was observed in the water for each of the microorganisms. However, the microbial count in chlorinated water after each treatment was below the detectable limit. This demonstrates that 100 ppm of chlorine is effective in significantly reducing pathogen populations and preventing cross-contamination. Even though chlorinated water remains a safe alternative for the decontamination of fresh produce and the prevention of cross-contamination, the combination of a physical treatment such as aeration with chemical treatments, applied as a hurdle technology, may improve the efficacy of sanitizers and allow for better removal of microorganisms from the surface of the produce during washing. While the findings of our study are promising, it is important to widen the application of forced aeration during produce washing by evaluating different fruits and vegetables and determining similarities or differences according to their surface characteristics. Additionally, the aeration system presented in our study is suitable for small and medium-scale production operations; therefore, this system can be upgraded in the future to determine the feasibility of applying a similar system to larger operations.

## Figures and Tables

**Figure 1 foods-13-00146-f001:**
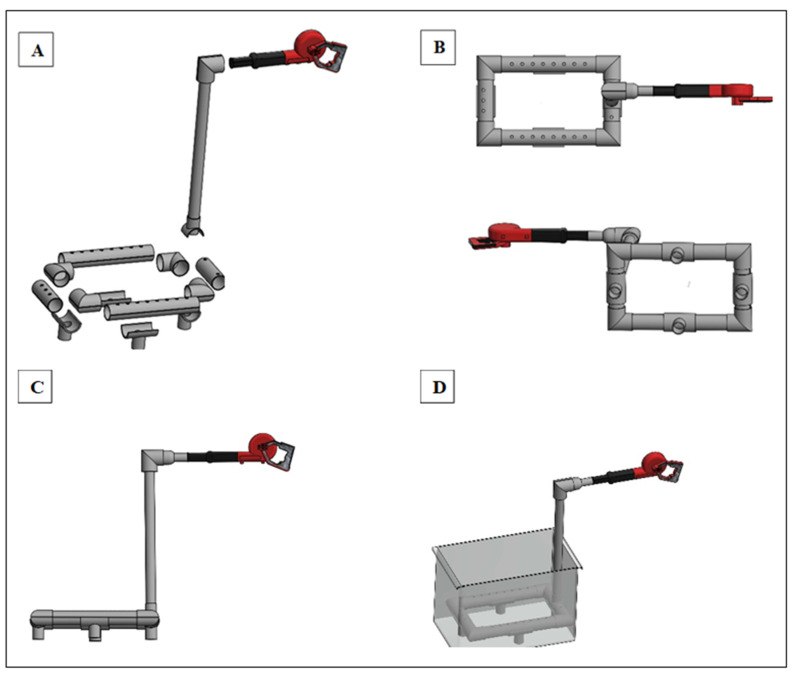
Aeration system design: (**A**) system assembly; (**B**) system top and bottom view; (**C**) lateral view of the system; (**D**) system placed inside the treatment container.

**Figure 2 foods-13-00146-f002:**
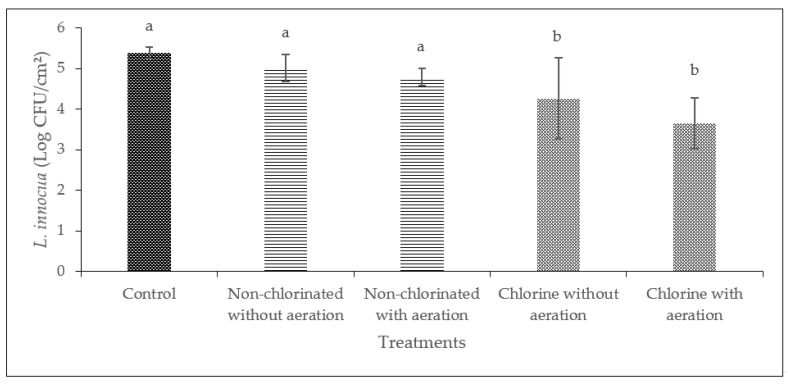
Effect of aeration during washing on cucumbers against *L. innocua*. (Different letters between bars indicate significant differences (*p* < 0.05)).

**Figure 3 foods-13-00146-f003:**
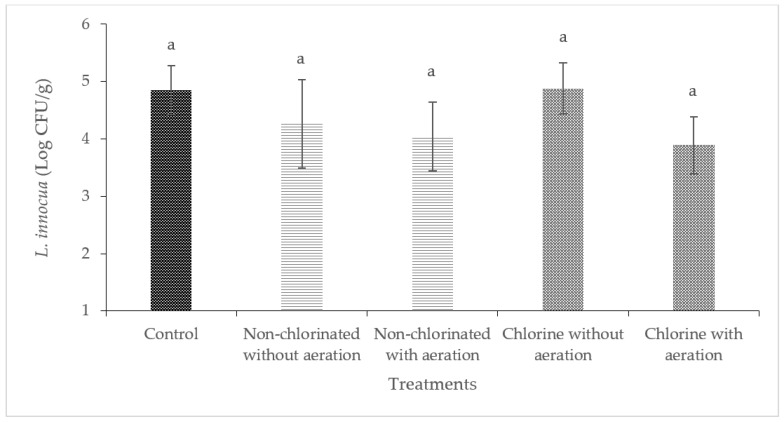
Effect of aeration during washing on bell peppers against *L. innocua*. (Different letters between bars indicate significant differences (*p* < 0.05)).

**Figure 4 foods-13-00146-f004:**
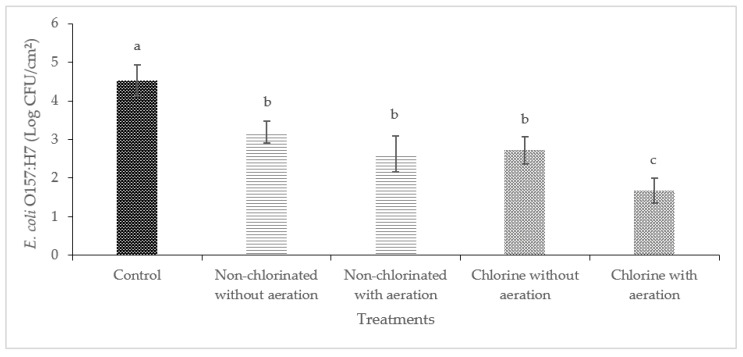
Effect of aeration during washing on cucumbers against *E. coli* O157:H7. (Different letters between bars indicate significant differences (*p* < 0.05)).

**Figure 5 foods-13-00146-f005:**
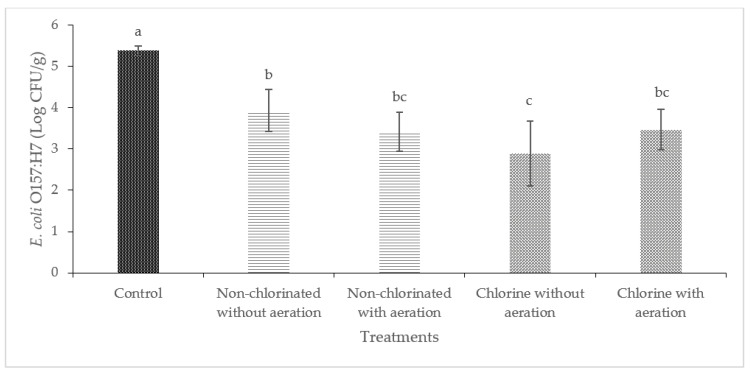
Effect of aeration during washing on bell peppers against *E. coli* O157:H7. (Different letters between bars indicate significant differences (*p* < 0.05)).

**Figure 6 foods-13-00146-f006:**
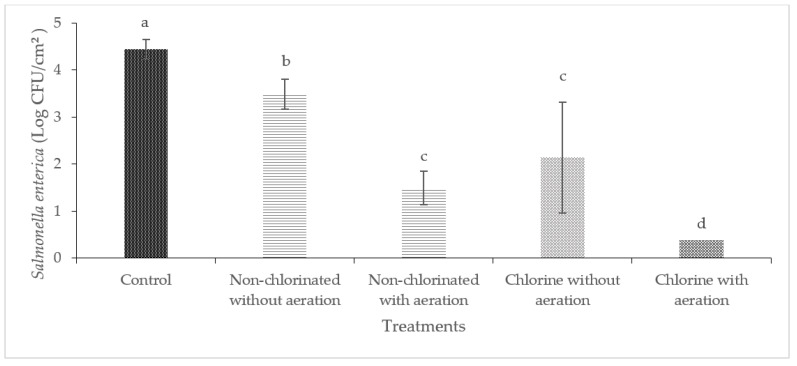
Effect of aeration during washing of cucumbers against *Salmonella enterica*. (Different letters between bars indicate significant differences (*p* < 0.05)).

**Figure 7 foods-13-00146-f007:**
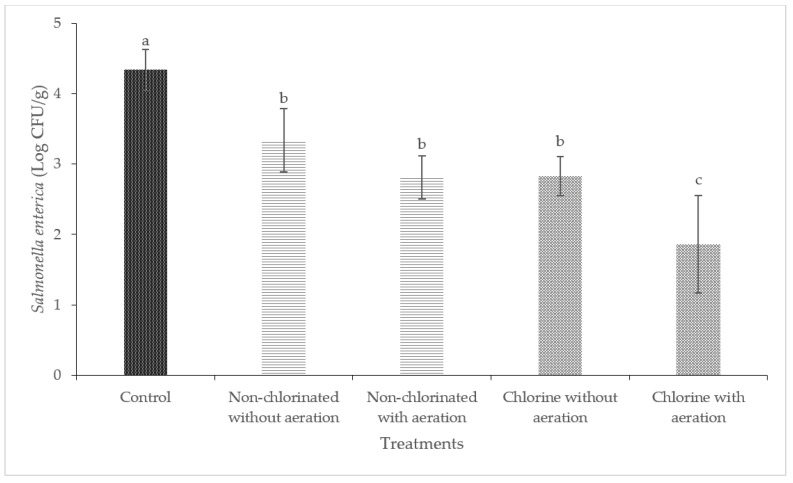
Effect of aeration during washing of bell peppers against *Salmonella enterica*. (Different letters between bars indicate significant differences (*p* < 0.05)).

**Table 1 foods-13-00146-t001:** Level of contamination in 1 mL of non-chlorinated wash water and chlorine solution expressed in log CFU/mL is presented as mean ± standard deviation.

Treatment ^*^	*L. innocua*		*E. coli* O157:H7		*Salmonella enterica*	
	Bell Peppers	Cucumbers	Bell Peppers	Cucumbers	Bell Peppers	Cucumbers
Non-chlorinated water without bubbles	1.41 ± 0.12 ^A^	2.20 ± 0.03 ^A^	2.21 ± 0.06 ^A^	<1.20 ^B^	1.90 ± 0.26 ^A^	2.00 ± 0.70 ^A^
Non-chlorinated water with bubbles	1.35 ± 0.60 ^A^	2.33 ± 0.11 ^A^	<1.10 ^B^	2.57 ± 0.14 ^A^	1.98 ± 0.66 ^A^	2.11 ± 1.09 ^A^
Chlorine without bubbles	<1.10 ^B^	<1.20 ^B^	<1.10 ^B^	<1.20 ^B^	<1.10 ^B^	<1.20 ^B^
Chlorine with bubbles	<1.10 ^B^	<1.20 ^B^	<1.10 ^B^	<1.20 ^B^	<1.10 ^B^	<1.20 ^B^

* The value of 1.10 log CFU/mL is the detectable limit of the microorganisms in non-chlorinated wash water and chlorinated solution when treating bell peppers. The value of 1.20 log CFU/mL is the detectable limit of the microorganisms in non-chlorinated wash water and chlorinated solution when treating cucumbers. (Different letters, A or B, in each produce between treatments represent significant differences (*p* < 0.05)).

## Data Availability

The data presented in this study are available within the article.

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
