# Peer review of "Evaluation of the Effectiveness of Aeration and Chlorination during Washing to Reduce E. coli O157:H7, Salmonella enterica, and L. innocua on Cucumbers and Bell Peppers"

_foods, 2023, doi:10.3390/foods13010146_

Round 1

Reviewer 1 Report

Comments and Suggestions for Authors

The manuscript titled "Evaluation of the Effectiveness of Aeration and Chlorination during Washing to Reduce Foodborne Pathogens on Cucumbers and Bell Peppers" presents updated findings on the role of aeration and chlorination in washing to remove microorganisms from cucumbers and bell peppers. This is an important topic due to the serious health risks associated with fresh produce as a significant source of foodborne illness outbreaks. With some necessary improvements, the manuscript can reach its full potential.

Following are the comments.

Add the effect of chlorination to the objectives of the study in lines 103-105.

Provide the CFU/ml bacterial load in 200 µl used for spot inoculation of each bell pepper and cucumber (Line 125).

Provide the weight of the used bell pepper and cucumber.

I strongly recommend providing a figure of the aeration system design.

The authors should have stored the vegetables for a few hours at 4 ÌŠC to allow bacterial attachment.
There seems to be some confusion regarding the analysis method (line 156) used. It is unclear why different amounts of PBS were used for the pepper and cucumber samples. Can you confirm if the homogenization step was carried out in an amount of diluent to obtain the first concentration or dilution (1/10), as it should have been? It's important that you have clarity on this matter.

It is crucial to emphasize the weight of each sample in this context, as it can greatly impact the accuracy and reliability of the results.

Also, it is essential that you clarify whether or not you have diluted the wash solution of the treatments. Furthermore, I noticed a discrepancy between the text and Table 1. Lines 167-169 state that the wash solution of the cucumber treatments was taken, while Table 1 displays results for pepper as well. It is crucial that you revise this section (line 156: sample analysis) immediately and make the necessary corrections. I expect this to be addressed promptly.

As stated in lines 236-237 and Figure 1, washing with non-chlorinated water significantly and effectively reduced the level of E. coli O157:H7. It is noteworthy that this pathogen was not detected in the wash water, as shown in Table 1 (<1.2 CFU). Therefore, an explanation is required to clarify the reason behind the absence of organisms in the wash water.

In lines 333-334 of the text, the authors suggested that the failure of bubbles to efficiently remove L. innocua in their study could be attributed to the presence of biofilm that keeps the cells embedded in the produce even after washing. Since biofilm is a strain trait, I would like to inquire if the strains used in the study were capable of producing biofilm?

Author Response

Reviewer #1:

The manuscript titled "Evaluation of the Effectiveness of Aeration and Chlorination during Washing to Reduce Foodborne Pathogens on Cucumbers and Bell Peppers" presents updated findings on the role of aeration and chlorination in washing to remove microorganisms from cucumbers and bell peppers. This is an important topic due to the serious health risks associated with fresh produce as a significant source of foodborne illness outbreaks. With some necessary improvements, the manuscript can reach its full potential.

Following are the comments.

Add the effect of chlorination to the objectives of the study in lines 103-105.

We have applied this change to the specified section.

Provide the CFU/ml bacterial load in 200 µl used for spot inoculation of each bell pepper and cucumber (Line 125).

We have added the bacterial load for the spot inoculation (8 log CFU/ml).

Provide the weight of the used bell pepper and cucumber.

The average weight of bell pepper samples was 191.20±17.41 g and the average weight of cucumber samples was 244.85±26.66 g. This has been added to the end of Section 2.5 Sample analysis.

I strongly recommend providing a figure of the aeration system design.

We appreciate the reviewer’s comment and in order to aid viewers in understanding how the system looked we have included images of the aeration system in the text as Figure 1.

The authors should have stored the vegetables for a few hours at 4 ÌŠC to allow bacterial attachment.

Thank you for the suggestion, we will keep this in mind for our future projects.

There seems to be some confusion regarding the analysis method (line 156) used. It is unclear why different amounts of PBS were used for the pepper and cucumber samples. Can you confirm if the homogenization step was carried out in an amount of diluent to obtain the first concentration or dilution (1/10), as it should have been? It's important that you have clarity on this matter.

We have included an explanation in this section in which we detail that cucumber samples had a higher volume of PBS due to these samples being longer than bell pepper samples. The additional volume was therefore sufficient to properly massage the surface of the cucumbers and dislodge bacteria.

Concerning the second part of the reviewer’s comment, when determining the initial dilution we used the weight of the sample and the diluent added to that sample (this has been clarified at the end of the sample analysis section). Afterward, the initial dilution factor was taken into consideration for the calculation of colony-forming units. The specific calculation for initial dilution is:

It is crucial to emphasize the weight of each sample in this context, as it can greatly impact the accuracy and reliability of the results.

We appreciate the reviewer’s suggestion and we have included clarification in the sample analysis section that the weight of each sample was indeed taken into consideration when determining initial dilutions and colony-forming units.

Also, it is essential that you clarify whether or not you have diluted the wash solution of the treatments. Furthermore, I noticed a discrepancy between the text and Table 1. Lines 167-169 state that the wash solution of the cucumber treatments was taken, while Table 1 displays results for pepper as well. It is crucial that you revise this section (line 156: sample analysis) immediately and make the necessary corrections. I expect this to be addressed promptly.

We have provided clarification on how the samples of wash solutions were processed. Indicating that from the 10 mL samples taken, 1 mL was removed and separated into three subsamples (300 µL, 300 µL, and 400 µL). These subsamples were then each spread plated onto separate petri dishes to get a count for 1 ml to examine low levels of microorganisms.

We apologize for the mistake, this was a typo in Lines 167-169, should also have mentioned that this procedure was done for the wash water of both cucumbers and bell peppers. We have amended the statement to read as follows:

Additionally, a 10 mL sample of the wash solution, both water and chlorine solution was immediately collected after washing the samples.

As stated in lines 236-237 and Figure 1, washing with non-chlorinated water significantly and effectively reduced the level of E. coli O157:H7. It is noteworthy that this pathogen was not detected in the wash water, as shown in Table 1 (<1.2 CFU). Therefore, an explanation is required to clarify the reason behind the absence of organisms in the wash water.

That’s a great point and an important result from this study. Several studies indicated that chlorine is very effective against bacterial pathogens when they directly interact in water as opposed to when they are attached to produce matrices. We added a few discussion points to justify the results.

In lines 333-334 of the text, the authors suggested that the failure of bubbles to efficiently remove L. innocua in their study could be attributed to the presence of biofilm that keeps the cells embedded in the produce even after washing. Since biofilm is a strain trait, I would like to inquire if the strains used in the study were capable of producing biofilm?

We would like to confirm that the L. innocua strain used (ATCC 33090) has been found to be capable of producing biofilms, as discussed in previously published articles such as:

https://doi.org/10.1111%2F1751-7915.13847

https://doi.org/10.4265/bio.20.153

Reviewer 2 Report

Comments and Suggestions for Authors

dear authors,

line 113; plz mention the company and the country where the media were produced

line 125: how do you reach this volume? it was better to count the bacteria in 1 ml and then calculate the volume 

line 155: why are the replication times different?

157: immediately after washing? when did you take the samples? plz mention

plz show the significant difference among groups in table 1

Comments on the Quality of English Language

Minor editing of English language required

Author Response

Reviewer #2:

line 113; plz mention the company and the country where the media were produced

We have applied the change as suggested by the reviewer.

line 125: how do you reach this volume? it was better to count the bacteria in 1 ml and then calculate the volume 

The volume was based on previous studies, we have included these as references in the mentioned section. We have also included here and throughout the manuscript clarification that the inoculum had an initial concentration of 8 log CFU/mL. Indeed, the microbial load was examined using 1 mL with several dilutions.

line 155: why are the replication times different?

Some variation in results was observed in one of the cucumber trials, so it was decided to include an additional replication of the trial.

157: immediately after washing? when did you take the samples? plz mention

We have changed the phrasing to mention washing to clarify this statement. Samples were removed after washing and placed in stomacher bags, along with diluent. Afterward, samples were homogenized and the liquid in the stomacher bags was serially diluted and plated in respective agars. We have edited this section to provide further clarity on when the samples were taken.

plz show the significant difference among groups in table 1

Done

Reviewer 3 Report

Comments and Suggestions for Authors

The manuscript "Evaluation of the effectiveness of aeration and chlorination during washing to reduce foodborne pathogens on cucumbers and bell peppers" has an interesting outcome. However, the manuscript needs to be improved.

The discussion needs to review and compare the data with the literature. Many grammatically problematic sentences were found throughout the manuscript, which must be checked and corrected precisely by English editing services.

1. L13: Escherichia coli (write full form for the first time)

2. L15, 180, 185, and so on: Check and makes correction throughout the manuscript (P < 0.05)

3. L 402, 414, 415, and so on: Accession dates missing

4. L55: Put reference

5. L77: Remove old references such as Ref. no 20. (Use references last 5 years, for example: https://doi.org/10.1080/00439339.2023.2163044)

6. L82: Salmonella enterica serovar Typhimurium

7. L84: Mention the reduction value in % or logarithmic form. It can’t be 90.3% to 99.00% log reduction.

8. L86: Remove comma (74%, and 86%)

9. L92: Use the abbreviated form after the first time (L. monocytogenes)

10. L113, 120, 141, and so on: mention the company name, city, country, etc for all chemicals and instruments. Check and correct throughout the manuscript.
11. L119:
Convert all rpm into “g” forces

12. L115-120: What do you mean by activation of organisms? The complete bacterial growing processing and working bacterial concentration should be written clearly. Please rewrite the complete methods of inoculum preparation.

13. L124: What was the concentration of used microorganisms here and why did you use that concentration? Mention it properly with references. Moreover, the size of the food coupons should be mentioned.

14. In Fig. 2: why the bacterial load has increased in bell peppers after chlorine treatment rather than the control? Explain with proper references

15. Many grammatically problematic sentences were found throughout the manuscript, which must be checked and corrected precisely by English editing services.

Comments on the Quality of English Language

Many grammatically problematic sentences were found throughout the manuscript, which must be checked and corrected precisely by English editing services.

Author Response

Reviewer #3:

The manuscript "Evaluation of the effectiveness of aeration and chlorination during washing to reduce foodborne pathogens on cucumbers and bell peppers" has an interesting outcome. However, the manuscript needs to be improved.

The discussion needs to review and compare the data with the literature. Many grammatically problematic sentences were found throughout the manuscript, which must be checked and corrected precisely by English editing services.

We appreciate the reviewer's suggestion. We updated the discussion section and revised the grammar throughout the manuscript.

  1. L13: Escherichia coli (write full form for the first time)

Done

  1. L15, 180, 185, and so on: Check and makes correction throughout the manuscript (< 0.05)

Done

  1. L 402, 414, 415, and so on: Accession dates missing

Done

  1. L55: Put reference

Reference has been added.

  1. L77: Remove old references such as Ref. no 20. (Use references last 5 years, for example: https://doi.org/10.1080/00439339.2023.2163044)

We have included the suggested reference. Additionally, we used several newer studies throughout the manuscript.

  1. L82: Salmonella entericaserovar Typhimurium

Done

  1. L84: Mention the reduction value in % or logarithmic form. It can’t be 90.3% to 99.00% log reduction.

The study in question is replaced with a recent publication, this has been updated in the text.

  1. L86: Remove comma (74%, and 86%)

The study in question is replaced with a recent publication, this has been updated in the text.

  1. L92: Use the abbreviated form after the first time (L. monocytogenes)

Done

  1. L113, 120, 141, and so on: mention the company name, city, country, etc for all chemicals and instruments. Check and correct throughout the manuscript.

We have applied this correction in the specified lines and throughout the manuscript.

  1. L119: Convert all rpm into “g” forces

Done

  1. L115-120: What do you mean by activation of organisms? The complete bacterial growing processing and working bacterial concentration should be written clearly. Please rewrite the complete methods of inoculum preparation.

The activation of microorganisms is done before the beginning of each experiment because the stock cultures are stored at -80ËšC in vials containing inoculated medium and glycerol. While the bacterial cells are latent, temperature below freezing can lead to bacterial cells becoming viable but non-culturable (VBNC), which can lead to bacterial populations lower than the intended ones for the research project. Therefore, we activate these microorganisms by transferring them to a new medium and allowing them to grow during incubation for three successive passes to ensure that the majority of the bacterial cells are indeed culturable.

We have added more information in this section to aid readers to understand what was done with the inoculum preparation.

  1. L124: What was the concentration of used microorganisms here and why did you use that concentration? Mention it properly with references. Moreover, the size of the food coupons should be mentioned.

We have included references that use similar spot inoculation techniques as we have done in this study, to justify the use of this volume for the spot inoculation.

  1. In Fig. 2: why the bacterial load has increased in bell peppers after chlorine treatment rather than the control? Explain with proper references

The difference in microbial load while using chlorine without aeration treatment (4.88 Log CFU/g) compared to the control (4.85 log CFU/g), was not significantly different. Our data indicated chlorine alone may not be effective against microorganisms when they are attached to produce surfaces.

  1. Many grammatically problematic sentences were found throughout the manuscript, which must be checked and corrected precisely by English editing services.

We appreciate the reviewer's suggestion, and we would like to confirm that we have revised the grammar throughout the manuscript.

Reviewer 4 Report

Comments and Suggestions for Authors

The study, "Evaluation of the Effectiveness of Aeration and Chlorination During Washing to Reduce Foodborne Pathogens on Cucumbers and Bell Peppers," investigates the impact of aeration and chlorination on the reduction of foodborne pathogens. While the research addresses an important aspect of food safety, there are several areas that require improvement to enhance the overall quality and impact of the study.

The title is concise and reflects the study's focus. However, it could be beneficial to specify the foodborne pathogens under investigation.

The abstract lacks a clear statement of the study's aim.

Revisit page 6 line 216 and make the statement clearer.

Discussion and Interpretation: The discussion section lacks depth in analysing and interpreting the results. what are the implications of the findings for the broader field of food safety?

Discuss the significance of observed trends, potential reasons for variations, and how these findings contribute to existing knowledge.

“Different letters between bars indicate significant differences (P<0.05)” should be in a bracket to ease readability and understanding for figures 1-6

What are the study’s limitations? Explicitly acknowledge and discuss the limitations of the study.

Author Response

Reviewer #4:

The study, "Evaluation of the Effectiveness of Aeration and Chlorination During Washing to Reduce Foodborne Pathogens on Cucumbers and Bell Peppers," investigates the impact of aeration and chlorination on the reduction of foodborne pathogens. While the research addresses an important aspect of food safety, there are several areas that require improvement to enhance the overall quality and impact of the study.

The title is concise and reflects the study's focus. However, it could be beneficial to specify the foodborne pathogens under investigation.

Thank you for the suggestion, to give readers more clarity we have included the names of the microorganisms used.

The abstract lacks a clear statement of the study's aim.

We have included the following statement in the abstract:

“The study aimed to determine the effectiveness of chlorination and aeration in the removal of pathogens from the surface of produce.”

Revisit page 6 line 216 and make the statement clearer.

We have revisited the statement as follows:

“Bacteria become firmly attached to the microstructures present on bell peppers, making it difficult to effectively clean and disinfect during washing.”

Discussion and Interpretation: The discussion section lacks depth in analyzing and interpreting the results. What are the implications of the findings for the broader field of food safety?

The discussion section is revised as suggested.

Discuss the significance of observed trends, potential reasons for variations, and how these findings contribute to existing knowledge.

The discussion section is revised as suggested.

“Different letters between bars indicate significant differences (P<0.05)” should be in a bracket to ease readability and understanding for figures 1-6

Done 

What are the study’s limitations? Explicitly acknowledge and discuss the limitations of the study.

We have added statements in our conclusions, concerning the limitations of our study and how it could be addressed in future studies.

Round 2

Reviewer 1 Report

Comments and Suggestions for Authors

The authors have addressed most of the comments on the manuscript, but some points still require immediate attention to make it suitable for publication. We urge the authors to take prompt action in addressing these issues and to ensure that the manuscript meets all necessary standards.

Line 173, Section 2.5. Sample analysis: There is a lack of methodology in this part, and it is important to clarify whether the pepper and cucumber samples were homogenized with PBS and the homogenate was then diluted, or if they were rinsed with PBS and then the rinse subjected to 10-fold serial dilution to enumerate the bacterial load.

Line 180-181: “After massaging or homogenization, the samples were diluted accordingly in 9 mL test tubes containing PBS”. The volume of the sample being diluted in 9 mL PBS is not specified. Please provide the necessary information to complete the procedure.

Lines 182-186: “The samples were then plated on Xylose-lysine Deoxycholate Agar (XLD) (Hardy Diagnostics, Santa Maria, CA, USA) for Salmonella enterica…..ect”. The inoculation volume for each medium used was not specified in the given text.

All the above-required data is necessary to achieve accurate results, especially regarding bacterial counts or loads.

In line 180, you mentioned that you made a 10-fold serial dilution to determine the bacterial load of the samples. However, in lines 196-197, you indicated that the initial dilutions were calculated based on the weight of the sample and the amount of diluent added. The average weight of bell pepper samples was 191.20±17.41 g, while for cucumber samples, the average weight was 244.85±26.66 g. In lines 174-175, you also mentioned that each bell pepper sample was homogenized using a Bagmixer® 400 blender in 100 mL of PBS 1X immediately after washing. It's important to note that by homogenizing 100 ml of PBS with an average of 191 g pepper, you cannot obtain a first dilution of 1/10. Consequently, the results obtained may be misleading.

Please note that there is an inconsistency in your methodology. In line 180, you mentioned that a 10-fold serial dilution was made to determine the bacterial load of the samples. Additionally, in lines 196-197, you indicated that the initial dilutions were calculated based on the weight of the sample and the amount of diluent added. As mentioned in lines 197-198, the average weight of the bell pepper samples was 191.20±17.41 g, while for cucumber samples, the average weight was 244.85±26.66 g. Moreover, in lines 174-175, you mentioned that each bell pepper sample was homogenized using a Bagmixer® 400 blender in 100 mL of PBS 1X immediately after washing and cucumber whole samples 176 were placed in sterile bags containing 200 mL of PBS (lines 176-177). However, homogenizing 100 ml of PBS with an average of 191 g pepper does not result in a first dilution of 1/10, and the same for cucumber samples which could lead to misleading results. I recommend that you review your methodology to ensure the accuracy of your findings.

How did you sample the control?  

It appears that there is a discrepancy between the bacterial count of the control and the number of bacteria used in the experiment. As per your manuscript (Lines 133-134), the initial bacterial load was 8 log CFU/mL of bacteria for each bell pepper and cucumber. However, the log bacterial count in the control for different bacterial strains ranged from less than 5 to less than 6 log CFU/mL. Can you explain this inconsistency?

In my previous review report, I mentioned that the text on lines 260-261 clearly states that washing cucumbers and bell peppers with non-chlorinated water resulted in a significant reduction of E. coli O157:H7 levels (P < 0.05), as shown in Figure 4 and Figure 5. However, it is important to note that this pathogen was not detected in the wash water of cucumber samples, as indicated in Table 1 (<1.2 CFU). Therefore, it is crucial that an explanation be provided to clarify why there were no organisms present in the wash water of non-chlorinated water. Your response did not answer my question. I had specifically asked for an explanation about non-chlorinated water, not chlorinated water. Please provide me with the information I requested.

Author Response

We would like to thank you for your efforts in reviewing our manuscript . Please see the attachment.

Reviewer 2 Report

Comments and Suggestions for Authors

the authors have corrected all the issues in the manuscript.

Author Response

Dear reviewer,

We feel great thanks for your review work. Thank you again for all your helpful comments and suggestions. 

Reviewer 3 Report

Comments and Suggestions for Authors

Manuscript should be accepted in the present form.

Author Response

(The authors gave the same response as above.)
